# A New Contraction-Type Mapping on a Vectorial Dislocated Metric Space over Topological Modules

Ion Marian Olaru 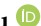

Department of Mathematics and Computer Science, Lucian Blaga University of Sibiu, Dr. I. Rațiu Street, No. 5–7, 550012 Sibiu, Romania; marian.olaru@ulbsibiu.ro

**Abstract:** One recent and prolific direction in the development of fixed point theory is to consider an operator $T : X \to X$ defined on a metric space $(X, d)$ which is an $F$—contraction, i.e., $T$ verifies a condition of type $\tau + F(d(T(x), T(y)) \leq F(d(x, y))$, for all $x, y \in X$, $T(x) \neq T(y)$, where $\tau > 0$ and $F : (0, \infty) \to \mathbb{R}$ satisfies some suitable conditions which ensure the existence and uniqueness for the fixed point of operator $T$. Moreover, the notion of F-contraction over a metric space $(X, d)$ was generalized by considering the notion of $(G, H)$—contraction, i.e., a condition of type $G(d(Tx, Ty)) \leq H(d(x, y))$, for all $x, y \in X$, $Tx \neq Ty$ for some appropriate $G, H : (0, \infty) \to \mathbb{R}$ functions. Recently, the abovementioned F-contraction theory was extended to the setup of cone metric space over the topological left modules. The principal objective of this paper is to introduce the concept of vectorial dislocated metric space over a topological left module and the notion of $A$-Cauchy sequence, as a generalization of the classical Cauchy sequence concept. Furthermore, based on the introduced concept, a fixed point result is provided for an operator $T : X \to X$, which satisfies the condition $(G, H)$—contraction, where $G, H$ are defined on the interior of a solid cone.

**Keywords:** vectorial dislocated metric spaces; topological left modules; fixed point theorems; generalized contraction

**MSC:** 47H09; 47H10

## 1. Introduction

A.I. Perov and his collaborators ([1–3]) presented the fixed point theory in K-metric and K-normal space. The general idea is the usage of an ordered Banach space, considered an alternative for the set of real numbers, as the codomain for a metric. To have a deeper analysis of the fixed point theory in K-metric and K-normed spaces, we guide the reader to [4]. Another important work is provided by Huang and Zhang [5], who presented this type of spaces as cone metric spaces, where the notion of convergent and Cauchy sequence was defined by using the solid cone notion, i.e., a cone with a nonempty interior. The authors also demonstrated some fixed point theorems in such spaces, further studies in fixed point results in cone metric spaces later being conducted. Another important result is the development of fixed point theory in ordered K-metric spaces or cone metric spaces provided by W.S. Du in [6]. It was shown that fixed point results in ordered K-metric spaces for map, fulfilling contractive conditions of a linear type in K-metric spaces, are treated as the corollaries of the matching theorems in metric spaces. In addition, in [7], another approach can be distinguished to demonstrate the equivalence between the vectorial version of fixed point results and the scalar one. I.M Olaru and N.A. Secelean, in [8] enlarged the abovementioned outcomes to a nonlinear contractive condition on TVS-cone metric space, further generalization being later identified. Liu and Xu [9] considered Banach algebra instead a Banach space, proposing the concept of cone metric space over the Banach algebra. The notions mentioned in [9] were broadened to the theory of cone metric space over the topological left modules in [10]. The authors of [10] extended the fixed

point results for linear contractions to cone metric spaces over topological left modules and built an example which showed that the abovementioned spaces cannot be metrizable. Another direction in the development of fixed point theory was given by Wardowski [11] in which the author introduced the notion of F-contraction over the scalar metric spaces. The abovementioned F-contraction theory was extended to the cone metric space over the topological left modules by A. Branga [12]. Furthermore, I.M Olaru and N.A. Secelean [13] generalized the notion of F-contraction, considering for an operator $T : X \to X$, a general contractive condition named $(G, H)-$contraction for some suitable $G, H$ functions. In this paper, we aim to extend the results from [13] to the setup of dislocated metric space over topological left modules. More specifically, our aim is to introduce the notion of $A$-Cauchy sequence which represents a generalization of the well-known Cauchy sequence definition. In addition, we introduce the notion of vectorial dislocated metric space as an extension of scalar dislocated metric space. Next, following the concepts from [10], we define a solid cone on a topological left module and by considering adequate condition for it we give a lemma used for proving the fact that the iteration sequence associated to an operator defined on a vectorial dislocated metric space is an A-Cauchy sequence. The main result of this paper is a fixed point result for an operator $T$ defined on a vectorial dislocated metric space which satisfies the condition (1).

$$G(d(Tx, Ty)) \leq H(d(x, y)), \quad (\forall) x, y \in X, Tx \neq Ty \tag{1}$$

## 2. Methods

In this section, concepts related to the topological ordered ring are presented. The reader can obtain more details from the work of Arnautov [14], Steinberg [15] and Warner [16].

**Definition 1.** *A ring $(R, +, \cdot)$ together with a partial order $\preceq$ is named partially ordered ring if:*

$(R_1)$ *$r_1 \preceq r_2$ entails $\alpha + r_1 + \beta \preceq \alpha + r_2 + \beta$, for all $r_1, r_2, \alpha, \beta \in R$.*
$(R_2)$ *$0 \preceq \alpha$ and $0 \preceq \beta$ entails $0 \preceq \alpha \cdot \beta$, for all $\alpha, \beta \in R$.*

Next we define $Positive(R) = \{\alpha \in R \mid 0 \preceq \alpha\}$, $U(R)$ the set of invertible elements of $R$ and $U(R) \cap Positive(R)$ will stands for $U_+(R)$.

**Definition 2.** *A ring $(R, +, \cdot)$, with $1_R \neq 0_R$ endowed with a topology $\tau_R$ is named a topological ring if the following maps are continuous:*

(i)   $R \times R \ni (r_1, r_2) \mapsto r_1 + r_2 \in R$;
(ii)  $R \ni r \mapsto -r \in R$;
(iii) $R \times R \ni (r_1, r_2) \mapsto r_1 \cdot r_2 \in R$.

*If $\tau_R$ is a Hausdorff topology, then $(R, +, \cdot, \tau_R)$ is named Hausdorff topological ring.*

**Definition 3.** *Let us consider $(R, +_R, \cdot_R)$ a ring. A left R-module is an abelian group $(E, +)$ with the external product $\cdot : R \times E \to E$, $(r, x) \ni R \times E \to r \cdot x \in E$ having the following properties:*

(i)   $(r_1 +_R r_2) \cdot x = r_1 \cdot x + r_2 \cdot x$, for all $r_1, r_2 \in R$, $x \in E$;
(ii)  $r \cdot (x_1 + x_2) = r \cdot x_1 + r \cdot x_2$, for all $r \in R$, $x_1, x_2 \in E$;
(iii) $1_R \cdot x = x$, for all $x \in E$;
(iv)  $(r_1 \cdot_R r_2) \cdot x = r_1 \cdot (r_2 \cdot x)$ for all $r_1, r_2 \in R$, $x \in E$.

**Definition 4.** *Let us consider $(R, +_R, \cdot_R)$ a ring. A right R-module is an abelian group $(E, +)$ with the external product $\cdot : E \times R \to E$, $(x, r) \ni E \times R \to x \cdot r \in E$ having the following properties:*

(i)   $x \cdot (r_1 +_R r_2) = x \cdot r_1 + x \cdot r_2$, for all $r_1, r_2 \in R$, $x \in E$;
(ii)  $(x_1 + x_2) \cdot r = x_1 \cdot r + x_2 \cdot r$, for all $r \in R$, $x_1, x_2 \in E$;
(iii) $x \cdot 1_R = x$, for all $x \in E$;
(iv)  $x \cdot (r_1 \cdot_R r_2) = (x \cdot r_1) \cdot r_2$ for all $r_1, r_2 \in R$, $x \in E$.

**Definition 5.** *Let us consider* $(R, +, \cdot, \tau_R)$ *a topological ring. A left R-module* $(E, +, \cdot)$ *endowed with a topology* $\tau_E$ *is called topological left* $R-module$ *if the maps*

(i)    $E \times E \ni (e_1, e_2) \mapsto e_1 + e_2 \in E$;
(ii)   $E \ni e \mapsto -e \in E$;
(iii)  $R \times E \ni (\alpha, x) \mapsto \alpha \cdot x \in E$,

*are continuous. A topological left R—module is denoted as* $(E, +, \cdot, \tau_E)$, *and in a simpler notation* $(E, \tau_E)$.

## 3. Results

**Definition 6.** *Let us consider* $(E, +, \cdot, \tau_E)$ *a topological left R-module. By cone, we understand a nonempty set* $P \subset E$ *which satisfies the next properties:*

$(P_1)$ *P is closed with respect to* $\tau_E$ *and* $P \neq \{0_E\}$;
$(P_2)$ $\alpha, \beta \in Positive(R)$ *and* $x, y \in P$ *entails* $\alpha \cdot x + \beta \cdot y \in P$;
$(P_3)$ $P \cap -P = \{0_E\}$.

*Moreover, if the interior of P, denoted by* $int(P)$, *is not empty the the cone P is named solid cone.*

Let us consider the cone $P \subset E$ and the partial order relation $\leq_P$ by

$$x \leq_P y \Longleftrightarrow y - x \in P. \tag{2}$$

In this paper, the notation $x <_P y$ will represent that $x \leq_P y$ but $x \neq y$, and $x \ll y$ indicates that $y - x \in int(P)$.

**Lemma 1.** *Let us consider* $(R, \oplus, \odot, \tau_R, \preceq)$ *a partially ordered topological ring, having identity* $1_R \in Positive(R)$, $(E, +, \cdot, \tau_E)$ *be a topological left R-module and* $P \subset E$ *be a solid cone E. The next conclusions hold:*

(i)    $P + P \subseteq P$;
(ii)   *if* $e_1 \leq_P e_2$ *and* $e_2 \ll_P e_3$, *then* $e_1 \ll_P e_3$, *for all* $e_1, e_2, e_3 \in E$;
(iii)  *if* $u_1 \leq_P v_1$ *and* $u_2 \leq_P v_2$, *then* $u_1 + u_2 \leq_P v_1 + v_2$, *for all* $u_1, u_2, v_1, v_2 \in E$.

**Proof.** (i)    It follows for $a = b = 1_R$ in Definition 6;
(ii)   It should be demonstrated that $e_3 - e_1 \in int(P)$ if $e_2 - e_1 \in P$ and $e_3 - e_2 \in int(P)$. Then we can find the neighborhood $V$ of $0_E$ with $e_3 - e_2 + V \subset P$. Consequently, $e_3 - e_1 + V = (e_3 - e_2) + V + (e_2 - e_1) \subset P + P \subset P$. Therefore $e_3 - e_1 \in int\, P$;
(iii)  Let us consider $u_1, u_2, v_1, v_2$ as in hypothesis (iii). Then $v_1 - u_1 \in P$ and $v_2 - u_2 \in P$. Taking into consideration the fact that $1_R \in Positive(R)$, we obtain further that $v_1 - u_1 + v_2 - u_2 \in P$ and consequently $u_1 + u_2 \leq_P v_1 + v_2$.
□

**Lemma 2.** *Let us suppose that* $(R, \oplus, \odot, \tau_R, \preceq)$ *is a partially ordered topological,* $(E, +, \cdot, \tau_E)$ *is a topological left R-module and* $P \subset E$ *a solid cone. The next conclusions hold:*

(i)    *if* $(R, \tau_R)$ *is a Hausdorff topological space,* $0_R \in Positive(R)'$, *where* $Positive(R)'$ *is derived set of* $Positive(R)$, $r \cdot int(P) \subseteq int(P)$ *for all* $r \in Positive(R) \setminus \{0_R\}$ *and* $x \in P, x \ll_P$ $c + c$ *for all* $c \in int(P)$, *then* $x = 0_E$;
(ii)   *if* $\{x_n\}_{n \in \mathbb{N}}, \{y_n\}_{n \in \mathbb{N}} \subset E$, $x, y \in E$, $x_n \xrightarrow{n} x$, $y_n \xrightarrow{n} y$, *and* $x_n \leq_P y_n$ *for all* $n \geq N_0$, *then* $x \leq_P y$.

**Proof.** (i)    Let us consider $c \in int(P)$. Due to the fact that $0_R \in Positive(R)'$ there is a sequence $(\alpha_n)_{n \in \mathbb{N}} \in \mathcal{R}_+ \setminus \{0_{\mathcal{R}}\}$ such that $\alpha_n \to 0_R$, as $n \to \infty$. Then $\alpha_n \cdot c \in int(P)$ and consequently $\alpha_n \cdot c + \alpha_n \cdot c - u \in int(P)$. Therefore, $\lim\limits_{n \to \infty} (\alpha_n \cdot c + \alpha_n \cdot c - u) = -u \in \overline{P} = P$. In this way, $u \in P \cap -P = \{0_E\}$.

(ii) Since $x_n \leq_P y_n$ for all $n \geq N_0$ we have that $y_n - x_n \in P$, $n \geq N$. By passing to limit as $n \to \infty$ and considering that $P$ is a closed set we have the conclusion. $\square$

**Definition 7.** *Let* $(R, \oplus, \odot, \tau_R, \preceq)$ *be a partially ordered topological,* $(E, +, \cdot, \tau_E)$ *be a topological left R-module, P be a solid cone of E and X be a nonempty set. By vectorial dislocated metric on X we understand a function* $d : X \times X \to E$ *that satisfies the following rules:*

$(d_1)$ $0_E \leq_P d(x, y)$ *for all* $x, y \in X$;
$(d_2)$ $d(x, y) \in Fr(P)$ *implies* $x = y$;
$(d_3)$ $d(x, y) = d(y, x)$ *for all* $x, y \in X$;
$(d_4)$ $d(x, y) \leq_P d(x, z) + d(z, y)$ *for all* $x, y, z \in X$.

*The pair* $(X, d)$ *will be named vectorial dislocated metric space over the topological left R-module. Moreover, if the condition* $(d_1)$, $(d_3)$, $(d_4)$ *are fulfilled and additionally*

$$(d_2')\ \ d(x, y) = 0_E \Longleftrightarrow x = y, \ \ (\forall) x, y \in X$$

*then d is a cone metric and the pair* $(X, d)$ *will be named cone metric space over the topological left R-module.*

**Example 1.** *Let us consider* $E = \prod_{m \in \mathbb{N}^*} \mathbb{R}$, $P = \prod_{m \in \mathbb{N}^*} \mathbb{R}_+$, $X = C([0, 1], \prod_{i \in \mathbb{N}} \mathbb{R}^n)$ *and* $d : X \times X \to P$ *expressed by*

$$d(x, y) = (d_i(x, y))_{i \in \mathbb{N}},$$

*where*

$$d_i(x, y) := \sup_{t \in [0, 1]} \|pr_i(x)(t) - pr_i(y)(t)\|_{\mathbb{R}^n} \cdot e^{-t}.$$

*Then* $(X, d)$ *is a vectorial dislocated metric space.*

**Proof.** $(d_1)$ Since $d_i(x, y) \in \mathbb{R}_+$, for all $x, y \in X$ it follows that $d(x, y) \in P$ i.e $0_E \leq_P d(x, y)$.
$(d_2)$ Let us assume that $d(x, y) \in Fr(P)$. It can be seen that there is $m_0 \in \mathbb{N}$ such that $d_{m_0}(x, y) = 0$. Consequently we obtain $x = y$.
$(d_3)$ It can be demonstrated using the fact that $d_i(x, y) = d_i(y, x)$ for all $x, y \in X$.
$(d_4)$ It can be proved using the fact that $d_i(x, y) \leq d_i(x, y) + d_i(y, z)$ for all $x, y, z \in X$. $\square$

**Example 2.** *Let us consider* $E = \mathbb{R}^2$, $P = \mathbb{R}_+^2$, $X = \mathbb{R}^2$ *and* $d : X \times X \to P$ *expressed by*

$$d(x, y) = (d_1(x, y), d_2(x, y))$$

*where:*

$$d_1(x, y) = \max\{\|x\|_{\mathbb{R}^2}, \|y\|_{\mathbb{R}^2}\}$$

$$d_2(x, y) = \begin{cases} 2 & , \quad x = y = (0, 0) \\ 1 & , \quad \quad otherwise \end{cases}$$

*Then* $(X, d)$ *is a vectorial dislocated metric space which is not cone metric space.*

**Proof.** $(d_1)$ Since $d_i(x, y) \in \mathbb{R}_+$, $i = \overline{1, 2}$ for all $x, y \in X$ it follows that $d(x, y) \in P$ i.e., $0_E \leq_P d(x, y)$.
$(d_2)$ Let us assume that $d(x, y) \in Fr(P)$. Then $d_1(x, y) = 0$ and consequently we obtain $x = y$.
$(d_3)$ It can be demonstrated using the fact that $d_i(x, y) = d_i(y, x)$ for all $x, y \in X$ and $i = \overline{1, 2}$.
$(d_4)$ It can be proved using the fact that $d_i(x, y) \leq d_i(x, y) + d_i(y, z)$ for all $x, y, z \in X$ and $i = \overline{1, 2}$.

Moreover we observe that for $x = y = (1,0)$ we have $d(x,y) = (1,1) \neq (0,0)$ and thus one has that $d$ is not a cone metric. $\square$

**Definition 8.** *Let $(R, \oplus, \odot, \tau_R, \preceq)$ be a partially ordered topological, $(E, +, \cdot, \tau_E)$ a topological left R-module, P a solid cone of E and $(X,d)$ a vectorial dislocated metric space over the topological left R-module.*

(1)  *A sequence $\{x_k\}_{k \in \mathbb{N}} \subset X$, satisfying the condition:*

     *for every $0_E \ll_P c$ there exists a number $k(c) \in \mathbb{N}$ such that for all $k \geq k(c)$ we have $d(x_k, x) \ll_P c$,*

    *is named convergent to a point $x \in X$;*

(2)  *A sequence $\{x_k\}_{k \in \mathbb{N}} \subset X$ is called as an $A-$Cauchy sequence if there exists a set $A \subseteq P + Fr(P)$ fulfilling the property: for every $c \in A + int(P)$ there exists a number $k(c) \in \mathbb{N}$ in order that $d(x_k, x_l) \ll_P c$, for all $k, l \geq k(c)$;*

(3)  *A sequence $\{x_k\}_{k \in \mathbb{N}} \subset X$ is called as an Cauchy sequence if for every $c \in int(P)$ there is a number $k(c) \in \mathbb{N}$ such that for all $k, l \geq n(c)$ we have $d(x_k, x_l) \ll_P c$.*

(4)  *The vectorial dislocated metric space $(X,d)$ is named $A$—complete if the following condition holds: any $A-$Cauchy sequence of points in X is convergent in X.*

**Remark 1.** *If $0_E \in Fr(P)$ we note that $\overline{P} = P = P - \{0_E\} \subseteq P - Fr(P)$. Hence $Fr(P) = \overline{P} \cap \mathcal{C}_E(int(P)) \subseteq (P - Fr(P)) \cap \mathcal{C}_E(int(P))$.*

**Remark 2.** *If $0_E \in Fr(P)$, then for $A = \{0_E\}$ we obtain the notion of Cauchy sequence.*

**Proof.** Since $0_E = 0_E + 0_E \in P + Fr(P)$ one has $A \subseteq P + Fr(P)$ and $A + int(P) = int(P)$. $\square$

**Definition 9.** *Let us consider $(R, \oplus, \odot, \tau_R, \preceq)$ a partially ordered topological, $(E, +, \cdot, \tau_E)$ a topological left R-module. A set $A \subset E$ is named bounded if for every neighborhood V of $0_E$ there is $\lambda_V \in U(Positive(R))$ in order that $A \subseteq \lambda_V \cdot V$.*

Next we make the following hypotheses:

**Hypotheses 1 (H1).** $P = \bigcup_{i \in \mathcal{I}} K_i$, *where*

(a)  $K_i \subseteq P$ *are sequentially compact subsets of $E$, for every $i \in \mathcal{I}$;*

(b)  *for every bounded sequence $\{x_n\}_{n \in \mathbb{N}} \subset P$ there exists $i_0 \in \mathcal{I}$ and $N(i_0) \in \mathbb{N}$ such that $x_n \in K_{i_0}$, for all $n \geq N(i_0)$.*

**Hypotheses 2 (H2).** *there exists the sets $B_j \subset E$, $j \in J$, such that for every $J_1 \subset J$ the family $(B_j)_{j \in J_1}$ is summable in E and*

(a)  $\sum_{j \in J_1} B_j \subseteq Fr(P)$;

(b)  $\sum_{j \in J_1} B_j + \sum_{j \in J_1} B_j + \sum_{j \in J_1} B_j + \sum_{j \in J_1} B_j \subseteq Fr(P)$.

**Hypotheses 3 (H3).** *$(R, \tau_R)$ is a Hausdorff topological space (i.e., any two distinct elements of $R$ can be separated by two disjoint neighbourhoods of them), $0_R \in Positive(R)'$, where $Positive(R)'$ is derived set of $Positive(R)$ and $r \cdot int(P) \subseteq int(P)$ for all $r \in Positive(R) \setminus \{0_R\}$.*

**Example 3.** *Let us consider $P = \prod_{m \in \mathbb{N}^*} \mathbb{R}_+$. Then*

(a)  *the Hypothesis (H1) is fulfilled for $K_i = \prod_{m \in \mathbb{N}^*} [0,i]$, $i \in \mathbb{N}^*$;*

(b)  *the Hypothesis (H2) is satisfied for $B_j = \{(0, \cdots, 0, x_j, 0, \cdots) \mid x_j \in \mathbb{R}_+\} \subset P$, $j \in \mathbb{N}^*$;*

(c)  *the Hypothesis (H3) is satisfied.*

We mention that the above hypotheses are necessary to prove the following lemma which represents the vectorial version of Lemma 1.1 pp.3 from [17]. It represents a useful instrument to prove that a sequence of elements from a dislocated metric space is an *A*—Cauchy sequence.

**Lemma 3.** *Let $\{x_n\}_{n\in\mathbb{N}}$ be a sequence in a vectorial dislocated metric space $(X, d)$ satisfying the properties:*

(i)  *the set $\{d(x_n, x_m) \mid n, m \in \mathbb{N}\}$ is bounded;*
(ii)  *the hypotheses $(H_1)$ and $(H_2)$ are fulfilled;*
(iii)  *$\tau_E$ is a Hausdorff topology;*
(iv)  *$0_E \in Fr(P)$.*

*Then:*

(1)  *if $\{x_n\}_{n\in\mathbb{N}}$ is not an $A-$Cauchy sequence, then there exists $c_0 \in A + int(P)$ and the subsequences $\{x_{m(k)}\}_{k\in\mathbb{N}}$, $\{x_{n(k)}\}_{k\in\mathbb{N}}$, checking for all $k \in \mathbb{N}$ the properties*

$$n(k) > m(k) > k, d(x_{m(k)}, x_{n(k)}) \not\ll_P c_0, \tag{3}$$

$$d(x_{m(k)}, x_{n(k)-1}) \ll_P c_0; \tag{4}$$

(2)  *in addition, if $\{x_n\}_{n\in\mathbb{N}}$ is such that $\lim_{n\to\infty} d(x_n, x_{n+1}) = z_0 \in \bigcup_{J_1 \subset J} (\sum_{j\in J_1} B_j)$, then there exist two elements $l \in int(P)$, $L \in int(P) - z_0 - z_0$ such that*

$$\lim_{k\to\infty} d(x_{m(k)}, x_{n(k)}) = l, \tag{5}$$

$$\lim_{k\to\infty} d(x_{m(k)-1}, x_{n(k)-1}) = L, \tag{6}$$

$$l - z_0 - z_0 \leq_P L \leq_P l + z_0 + z_0. \tag{7}$$

**Proof.** (1) Assuming that $\{x_n\}_{n\in\mathbb{N}}$ is not an $A$—Cauchy sequence. Then, we can point out $c_0 \in A + int(P)$ and the subsequences $\{x_{m_1(k)}\}_{k\in\mathbb{N}}$, $\{x_{n_1(k)}\}_{k\in\mathbb{N}}$, in order that $n_1(k) > m_1(k) > k$ and $c_0 - d(x_{m_1(k)}, x_{n_1(k)}) \notin int(P)$ for all $k \in \mathbb{N}$. Furthermore, for every $k \in \mathbb{N}$, corresponding to $m_1(k)$, we can take $n_1(k)$ to be the minimum integer with $n_1(k) > m_1(k)$ and $d(x_{m_1(k)}, x_{n_1(k)}) \not\ll_P c_0$, therefore $d(x_{m_1(k)}, x_{n_1(k)-1}) \ll_P c_0$.

According to the hypothesis $(H_1)$ applied for $\{d(x_{m_1(k)}, x_{n_1(k)})\}_{k\in\mathbb{N}}$, there is $i_0 \in \mathcal{I}$ and $N(i_0) \in \mathbb{N}$ in order that $d(x_{m_1(k)}, x_{n_1(k)}) \in K_{i_0}$, for all $k \geq N(i_0)$. Since $K_{i_0}$ is a sequentially compact set, it can be seen that there is a subsequence of $\{d(x_{m_1(k)}, x_{n_1(k)})\}_{k\in\mathbb{N}}$ which converges to a point $l \in K_{i_0} \subseteq P$. Therefore, a strictly increasing function $r : \mathbb{N} \to \mathbb{N}$ is obtained verifying for all $k \in \mathbb{N}$ we have $n_1(r(k)) > m_1(r(k)) > k$, $r(k) > k$ and

$$l = \lim_{k\to\infty} d(x_{m_1(r(k))}, x_{n_1(r(k))}).$$

By using similar arguments as the abovementioned applied to $\{d(x_{m_1(r(k))-1}, x_{n_1(r(k))-1})\}_{k\in\mathbb{N}}$, a strictly increasing function $s : \mathbb{N} \to \mathbb{N}$ is obtained, verifying for all $k \in \mathbb{N}$ the properties $n_1(r(s(k))) > m_1(r(s(k))) > k$, $s(k) > k$ and a point $L \in P$ such that

$$L = \lim_{k\to\infty} d(x_{m_1(r(s(k)))-1}, x_{n_1(r(s(k)))-1})$$

Consequently, the properties (3) and (4) are verified for $m, n : \mathbb{N} \to \mathbb{N}$, $m(k) = m_1(r(s(k)))$, $n(k) = n_1(r(s(k)))$, which are strictly increasing functions with $n(k) > m(k) > k$. Furthermore, we obtained that

$$l = \lim_{k\to\infty} d(x_{m_1(r(k))}, x_{n_1(r(k))}) = \lim_{k\to\infty} d(x_{m_1(r(s(k)))}, x_{n_1(r(s(k)))}) = \lim_{k\to\infty} d(x_{m(k)}, x_{n(k)}) \tag{8}$$

and

$$L = \lim_{k \to \infty} d(x_{m_1(r(s(k)))-1}, x_{n_1(r(s(k)))-1}) = \lim_{k \to \infty} d(x_{m(k)-1}, x_{n(k)-1}). \tag{9}$$

(2) Next, to prove the relation (5), we remark that (3) implies $c_0 - d(x_{m(k)}, x_{n(k)}) \notin int(P)$, for all $k \in \mathbb{N}$ and thus $c_0 - d(x_{m(k)}, x_{n(k)}) \in \mathcal{C}_E(int(P))$. For $k \to \infty$ via relation (8) we find

$$c_0 - l \in \overline{\mathcal{C}_E(int(P))} = \mathcal{C}_E(int(P)). \tag{10}$$

On the other side, we have that, for each $k \in \mathbb{N}$,

$$d(x_{m_1(r(k))}, x_{n_1(r(k))}) \leq_P d(x_{m_1(r(k))}, x_{n_1(r(k))-1}) + d(x_{n_1(r(k))-1}, x_{n_1(r(k))}) \ll_P$$

$$c_0 + d(x_{n_1(r(k))-1}, x_{n_1(r(k))}),$$

thus

$$c_0 - d(x_{m_1(r(k))}, x_{n_1(r(k))}) + d(x_{n_1(r(k))-1}, x_{n_1(r(k))}) \in int(P).$$

Considering the relation (8) and the hypothesis $\lim_{n \to \infty} d(x_n, x_{n+1}) = z_0$ and passing to the limit as $k \to \infty$ in the previous relation, we deduce

$$c_0 - l + z_0 \in \overline{int(P)} \subseteq P.$$

Since $z_0 \in \bigcup_{J_1 \subset J} (\sum_{j \in J_1} B_j)$, it can be seen that there is $J_1 \subset J$ such that $z_0 \in \sum_{j \in J_1} B_j$. Taking into account the hypothesis $(H_2)(a)$, we obtain $z_0 \in Fr(P)$. Therefore,

$$c_0 - l \in P - z_0 \subseteq P - Fr(P). \tag{11}$$

Consequently, from the relations (10) and (11) we obtain

$$c_0 - l \in (P - Fr(P)) \cap \mathcal{C}_E(int(P)).$$

We deduce that

$$l \in c_0 - (P - Fr(P)) \cap \mathcal{C}_E(int(P))$$
$$\subseteq A + int(P) - (P - Fr(P)) \cap \mathcal{C}_E(int(P)). \tag{12}$$

Since $A \subseteq P + Fr(P)$ and $Fr(P) \subseteq (P - Fr(P)) \cap \mathcal{C}_E(int(P))$, we obtain

$$A \subseteq P + (P - Fr(P)) \cap \mathcal{C}_E(int(P)),$$

hence

$$A - (P - Fr(P)) \cap \mathcal{C}_E(int(P)) \subseteq P.$$

Consequently,

$$l \in int(P) + P \subseteq int(P). \tag{13}$$

Furthermore, by applying the triangular inequality, it is obtained that

$$d(x_{m_1(m_2(m_3(k)))}, x_{n_1(n_2(n_3(k)))}) \leq_P$$

$$d(x_{m_1(m_2(m_3(k)))}, x_{m_1(m_2(m_3(k)))-1}) + d(x_{m_1(m_2(m_3(k)))-1}, x_{n_1(n_2(n_3(k)))})$$

$$\leq_P d(x_{m_1(m_2(m_3(k)))}, x_{m_1(m_2(m_3(k)))-1})$$

$$+d(x_{m_1(m_2(m_3(k)))-1}, x_{n_1(n_2(n_3(k)))-1}) + d(x_{n_1(n_2(n_3(k)))-1}, x_{n_1(n_2(n_3(k)))})$$

$$d(x_{m_1(m_2(m_3(k)))}, x_{n_1(n_2(n_3(k)))}) - d(x_{m_1(m_2(m_3(k)))-1}, x_{m_1(m_2(m_3(k)))})$$

$$-d(x_{n_1(n_2(n_3(k)))-1}, x_{n_1(n_2(n_3(k)))}) \leq_P d(x_{m_1(m_2(m_3(k)))-1}, x_{n_1(n_2(n_3(k)))-1})$$

and

$$d(x_{m_1(m_2(m_3(k)))-1}, x_{n_1(n_2(n_3(k)))-1}) \leq_P d(x_{m_1(m_2(m_3(k)))}, x_{n_1(n_2(n_3(k)))})$$

$$+d(x_{m_1(m_2(m_3(k)))-1}, x_{m_1(m_2(m_3(k)))}) + d(x_{n_1(n_2(n_3(k)))-1}, x_{n_1(n_2(n_3(k)))}).$$

From the above inequalities, we infer

$$l - z_0 - z_0 \leq_P L \leq_P l + z_0 + z_0.$$

As $l \in int(P)$, we have $0_E \ll_P l$. The last inequalities lead us to

$$0_E \ll_P l \leq_P L + z_0 + z_0,$$

hence

$$L + z_0 + z_0 \in int(P),$$

thus

$$L \in int(P) - z_0 - z_0.$$

$\square$

**Definition 10.** *Let us consider* $(R, \oplus, \odot, \tau_R, \preceq)$ *a partially ordered topological ring and* $(E, +, \cdot, \tau_E)$ *a topological left R-modul,* $P \subset E$ *a solid cone.* $\mathcal{G}$ *is defined as the set of all pairs of mappings* $G, H : int(P) \to E$ *which fulfill the following conditions:*

$(C_1)$ *$G$ and $H$ are sequentially continuous on* $int(P)$;

$(C_2)$ *if* $\{d_n\}_{n \in \mathbb{N}} \subset P$ *is such that* $d_{n+1} \ll d_n$ *for all* $n \in \mathbb{N}$ *and for every* $c \in int(P)$ *there is a number* $N(c) \in \mathbb{N}$ *in order that for all* $n \geq N(c)$ *we have* $G(d_n) + c \ll_P 0_E$, *then* $d_n \to z_0 \in \bigcup_{J_1 \subset J} (\sum_{j \in J_1} B_j)$;

$(C_3)$ *for every* $r, t \in P, r \neq t$, *satisfying* $G(r) \leq_P H(t)$, *we have* $r \ll_P t$;

$(C_4)$ *for every* $r, t \in P, r \leq_P t$, *we have* $H(r) \leq_P H(t)$;

$(C_5)$ *if* $\{d_n\}_{n \in \mathbb{N}} \subset P$ *is such that* $d_{n+1} \ll d_n$ *for all* $n \in \mathbb{N}$ *and* $c \in int(P)$ *then there is* $N(c) \in \mathbb{N}$ *in order that* $G(d_0) + \sum_{k=1}^{n} (H(d_{k-1}) - G(d_{k-1})) + c \ll_P 0_E$, *for every* $n \geq N(c)$.

**Theorem 1.** *Let* $(R, \oplus, \odot, \tau_R, \preceq)$ *be a partially ordered topological ring* $(E, +, \cdot, \tau_E)$ *a Hausdorff topological left R-modul,* $P \subset E$ *a solid cone,* $(X, d)$ *an* $A-$ *complete vectorial dislocated metric space and* $T : X \to X$ *such that*

*(i)     the hypothesis* $(H_1)$, $(H_2)$ *and* $(H_3)$ *hold;*

*(ii)    $D = \{d(T(x), T(y)) \mid x, y \in X\}$ is bounded;*

*(iii)   $0_E \in Fr(P)$;*

*(iv)    there exists* $(G, H) \in \mathcal{G}$ *in order that*

$$G(d(Tx, Ty)) \leq_P H(d(x, y))), \quad (\forall)x, y \in X, d(Tx, Ty) \in int(P). \tag{14}$$

*Then $T$ has an unique fixed point* $x^\star \in X$, *and for every* $x_0 \in X$ *the sequence* $\{T^n x_0\}_{n \in \mathbb{N}}$ *is convergent to* $x^\star \in X$.

**Proof.** In the first place, we remark that the condition (14) leads us to the fact that $T$ has at most one fixed point. Indeed, if $x_1^\star, x_2^\star \in X$ is in order that $Tx_1^\star = x_1^\star \neq x_2^\star = Tx_2^\star$, then using relation (14) we find

$$G(d(Tx_1^\star, Tx_2^\star)) \leq_P H(d(x_1^\star, x_2^\star)),$$

hence

$$G(d(x_1^\star, x_2^\star)) \leq_P H(d(x_1^\star, x_2^\star)),$$

thus $d(x_1^\star, x_2^\star) \ll d(x_1^\star, x_2^\star)$, which is in contradiction with $0_E \in Fr(P)$. Therefore, $x_1^\star = x_2^\star$, i.e., $T$ has at most one fixed point.

To demonstrate that $T$ has a fixed point let $x_0 \in X$ be an arbitrary point. We define a sequence $\{x_n\}_{n \in \mathbb{N}}$ by $x_n = Tx_{n-1}, n \geq 1$ and let denote $d_n = d(x_{n+1}, x_n) \in P, n \in \mathbb{N}$.

If there is $n_0 \in \mathbb{N}$ in order that $d(x_{n_0}, x_{n_0+1}) \in Fr(P)$ then $x_{n_0+1} = x_{n_0}$ and therefore $x_{n_0}$ is a fixed point of $T$. Next, we suppose that $d(x_{n+1}, x_n) \in int(P)$ for all $n \in \mathbb{N}$. The relation (14) implies that $G(d_n) \leq_P H(d_{n-1})$ for all $n \geq 1$ and consequently $d_n \ll d_{n-1}$ for all $n \geq 1$. From the previous inequality we obtain that

$$\sum_{k=1}^{n}(G(d_k) - G(d_{k-1})) \leq_P \sum_{k=1}^{n}(H(d_{k-1}) - G(d_{k-1}))$$

so

$$G(d_n) \leq_P G(d_0) + \sum_{k=1}^{n}(H(d_{k-1}) - G(d_{k-1})),$$

for every $n \geq 1$. Let us consider $c \in int(P)$ is an arbitrary element. From condition $(C_5)$ we deduce that there is $N(c) \in \mathbb{N}$ in order that

$$G(d_n)) + c \leq_P G(d_0) + \sum_{k=1}^{n}(H(d_{k-1}) - G(d_{k-1})) + c \ll_P 0_E,$$

for every $n \geq N(c)$. Hence, $G(d_n)) + c \ll_P 0_E$ for every $n \geq N(c)$ and via condition $(C_2)$ we find that there exists an element $z_0 \in \bigcup_{J_1 \subset J} (\sum_{j \in J_1} B_j)$ such that $d_n \to z_0$.

Now, we assume that $\{x_n\}_{n \in \mathbb{N}}$ is not an $A-$Cauchy sequence. According to Lemma 3, we can obtain two subsequences $\{x_{m(k)}\}_{k \in \mathbb{N}}$, $\{x_{n(k)}\}_{k \in \mathbb{N}}$ and two elements $l \in int(P)$, $L \in int(P) - z_0 - z_0$ such that

$$\lim_{k \to \infty} d(x_{m(k)}, x_{n(k)}) = l,$$

$$\lim_{k \to \infty} d(x_{m(k)-1}, x_{n(k)-1}) = L,$$

$$l - z_0 - z_0 \leq_P L \leq_P l + z_0 + z_0.$$

Since $l \in int(P)$ and $\lim_{k \to \infty} d(x_{m(k)}, x_{n(k)}) = l$, we deduce that there is $K \in \mathbb{N}$ in order that $d(x_{m(k)}, x_{n(k)}) \in int(P)$, for all $k \geq K$. Via relation (14), hypothesis $(H_2)(a)$ and condition $(C_4)$ it follows that

$$G(d(x_{m(k)}, x_{n(k)})) \leq_P H(d(x_{m(k)-1}, x_{n(k)-1})) \leq_P H(d(x_{m(k)-1}, x_{n(k)-1}) + z_0 + z_0),$$

for every $k \geq K$. As $G$ and $H$ are sequentially continuous on $int(P)$ from the last inequality we obtain

$$G(l) \leq_P H(L + z_0 + z_0).$$

By using condition $(C_3)$, we obtain $l \ll_P L + z_0 + z_0$. Considering the relation $L \leq_P l + z_0 + z_0$, from the previous inequality we find $l \ll_P l + z_0 + z_0 + z_0 + z_0$, hence $z_0 + z_0 + z_0 + z_0 \in int(P)$. On the other hand, $z_0 \in \bigcup_{J_1 \subset J} (\sum_{j \in J_1} B_j)$, thus there exists $J_1 \subset J$ such that $z_0 \in \sum_{j \in J_1} B_j$. Therefore, $z_0 + z_0 + z_0 + z_0 \in \sum_{j \in J_1} B_j + \sum_{j \in J_1} B_j + \sum_{j \in J_1} B_j + \sum_{j \in J_1} B_j$ and using Hypothesis (H2)$(b)$ it follows that $z_0 + z_0 + z_0 + z_0 \in Fr(P)$. Consequently, $z_0 + z_0 + z_0 + z_0 \in int(P) \cap Fr(P) = \emptyset$, which is a contradiction. Hence, $\{x_n\}_{n \in \mathbb{N}}$ is an $A$—Cauchy sequence and from the $A$—completeness of $X$ there exists $x^\star \in X$ such that $x_n \to x^\star$ as $n \to \infty$.

Next we prove that $Tx^\star = x^\star$. Arguing by contradiction, let us suppose that $Tx^\star \neq x^\star$. We define the set $B = \{n \in \mathbb{N} \mid x_n = Tx^\star\}$. There are two cases relative to the set $B$. In the first case, if $B$ is not a finite set, then we can find $\{x_{n(k)}\}_{k \in \mathbb{N}}$ of $\{x_n\}_{n \in \mathbb{N}}$ which converges to $Tx^\star$. However, $x_n \to x^\star$ as $n \to \infty$ and the uniqueness of the limit leads us to $Tx^\star = x^\star$. In the second case, if $B$ is a finite set, then $d(x_n, Tx^\star) \in int(P)$, for infinitely many $n \in \mathbb{N}$.

Hence, there is a subsequence $\{x_{m(k)}\}_{k \in \mathbb{N}}$ of $\{x_n\}_{n \in \mathbb{N}}$ such that $d(x_{m(k)}, Tx^\star) \in int(P)$, for all $k \in \mathbb{N}$. Using relation (14), we obtain

$$G(d(Tx^\star, x_{m(k)})) \leq_P H(d(x^\star, x_{m(k)-1})) \text{ for all } k \in \mathbb{N}.$$

Taking into account the condition $(C_3)$, we deduce

$$d(Tx^\star, x_{m(k)}) \ll_P d(x^\star, x_{m(k)-1}).$$

By using the previous relation and the triangle inequality for the vectorial dislocated metric $d$, we find

$$d(Tx^\star, x^\star) \leq_P d(Tx^\star, x_{m(k)}) + d(x_{m(k)}, x^\star)$$

$$\ll_P d(x^\star, x_{m(k)-1}) + d(x_{m(k)}, x^\star) \text{ for all } k \in \mathbb{N}. \tag{15}$$

We select $c \in int(P)$ be an arbitrary element. Because $x_n \to x^\star$, for $n \to \infty$ we have $d(x_n, x^\star) \ll_P c$, for any $n \geq N(c)$. Since $m(k) > k$ for all $k \in \mathbb{N}$, it follows that $d(x_{m(k)}, x^\star) \ll_P c$, $d(x_{m(k)-1}, x^\star) \ll_P c$ for all $k \geq N(c) + 1$. For this reason,

$$d(x_{m(k)}, x^\star) + d(x_{m(k)-1}, x^\star) \ll_P c + c \text{ for all } k \geq N(c) + 1. \tag{16}$$

From the inequalities (15) and (16), we obtain

$$d(Tx^\star, x^\star) \ll_P c + c \text{ for all } c \in int(P).$$

Considering the hypothesis $(H_3)$ and utilizing Lemma 2 (i), it can be deduced that $d(Tx^\star, x^\star) = 0_E \in Fr(P)$, hence $Tx^\star = x^\star$, so $x^\star$ is a fixed point of $T$. □

**Example 4.** *Let us consider* $X = C([0,1], \prod_{i \in \mathbb{N}} \mathbb{R}^n)$ *, the vectorial dislocated metric d defined as in Example 1 and the following integral*

$$x(t) = f(t) + \int_0^t K(t, s, x(s)) ds, \ t \in [0,1] \tag{17}$$

*where*

(i)　$f \in C([0,1], \prod_{i \in \mathbb{N}} \mathbb{R}^n), K \in C([0,1] \times [0,1] \times \prod_{i \in \mathbb{N}} \mathbb{R}^n), \prod_{i \in \mathbb{N}} \mathbb{R}^n));$

(ii)　*there exists* $\tau_i > 0$ *such that*

$$\|pr_i(K(t, s, x(s))) - pr_i(K(t, s, y(s)))\|_{\mathbb{R}^n} \leq \frac{\|pr_i(x(s)) - pr_i(y(s))\|_{\mathbb{R}^n}}{\tau_i \cdot d_i(x, y) + 1},$$

　　*for each* $i \in \mathbb{N}$, $x, y \in C([0,1], \prod_{i \in \mathbb{N}} \mathbb{R}^n).$

*Then Equation (17) has a unique solution in* $C([0,1], \prod_{i \in \mathbb{N}} \mathbb{R}^n).$

**Proof.** Let us consider

$$T : X \to X,$$

$$T(x)(t) = f(t) + \int_0^t K(t, s, x(s)) ds$$

and $G, H : int(P) \to E$ defined by

$$G(a_1, a_2, \cdots, a_i, \cdots) = (\tau_1 - \frac{1}{a_1}, \tau_2 - \frac{1}{a_2}, \cdots, \tau_i - \frac{1}{a_i}, \cdots)$$

$$H(a_1, a_2, \cdots, a_i, \cdots) = (-\frac{1}{a_1}, -\frac{1}{a_2}, \cdots, -\frac{1}{a_i}, \cdots),$$

where $a_i > 0$ for all $i \in \mathbb{N}$. We remark that the pair $(G, H) \in \mathcal{G}$ and for each $x, y \in X$ we have

$$Gd(T(x), T(y))) \leq_P H(d(x, y)). \tag{18}$$

Indeed Equation (18) is equivalent with

$$\tau_i - \frac{1}{d_i(T(x), T(y))} < -\frac{1}{d_i(x, y)}. \tag{19}$$

The relation (19) can be obtained taking into account that for all $x, y \in X$ and $t \in [0, 1]$ we have:

$$\|pr_i(T(x))(t) - pr_i(T(y))(t)\|_{\mathbb{R}^n} \leq \int_0^t \|pr_i(K(t, s, x(s))) - pr_i(K(t, s, y(s)))\|_{\mathbb{R}^n} ds \leq$$

$$\frac{1}{\tau_i \cdot d_i(x, y) + 1} \cdot \int_0^t \|pr_i(x(s)) - pr_i(y(s))\|_{\mathbb{R}^n} ds \leq \frac{d_i(x, y)}{\tau_i \cdot d_i(x, y) + 1} \cdot e^t.$$

Therefore

$$d_i(T(x), T(y)) \leq \frac{d_i(x, y)}{\tau_i \cdot d_i(x, y) + 1}$$

and thus relation (19). □

### 4. Conclusions

In this paper we have achieved the following:

- Introduced the notion of vectorial dislocated metric space on a topological left module as an extension of scalar dislocated metric space. This concept generalizes the concept of cone metric space over a topological left module from [10];
- Introduced the notion of A-Cauchy sequence which represents a generalization of the well-known Cauchy sequence definition;
- By using the notion of a solid cone on a topological left module and considering an adequate condition for it, we give a lemma used for proving the fact that the iteration sequence associated to an operator defined on a vectorial dislocated metric space is an A-Cauchy sequence. In this way, we generalized the results from [17];
- As a main result of this paper, we provided a fixed point result for a self operator $T$ defined on a vectorial dislocated metric space which satisfies the condition (1). It represents a generalization of results from [12];
- There was an application of the main result given to the existence and uniqueness of the solution for a vectorial integral equation.

As further research direction we can mention the following ones:

- To adapt the main results to an iterative system function with application to fractals theory;
- To realize a study related with data dependence: continuity and smooth dependence for the fixed point of the operator $T$;
- To simplify the condition imposed to the solid cone.

**Author Contributions:** Conceptualization, I.M.O.; methodology, I.M.O.; formal analysis, I.M.O.; writing—original draft preparation, I.M.O.; writing—review and editing, I.M.O.; funding acquisition, I.M.O. All authors have read and agreed to the published version of the manuscript.

**Funding:** This research was funded by Lucian Blaga University of Sibiu through the research grant LBUS-IRG-2022-08.

**Institutional Review Board Statement:** Not applicable.

**Acknowledgments:** The authors thank the anonymous reviewers for their valuable comments and suggestions which helped us to improve the content of this paper.

**Conflicts of Interest:** The author declares no conflict of interest.

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
