# Peer review of "A New Contraction-Type Mapping on a Vectorial Dislocated Metric Space over Topological Modules"

_axioms, doi:10.3390/axioms11080405_

Round 1

Reviewer 1 Report

From the study of this paper I have come to the following conclusions:

  • The title of the paper is adequate
  • The abstract should be rewritten to represent the previous work and why the authors studied this topic.
  • Quoting a set of publications (e.g. “  [9], [13],[14]” . "[1], [2],[7] ......[18]"  In this form is unacceptable. Author should analyze the literature more selectively and indicate the most important publications in the analyzed issue together with a commentary.
  • Highlight some applications of the proposed contraction.
  • Concluding points can be made more focusing.
  • No recent citations and references are being cited in this manuscript and this is not  acceptable.

Author Response

Reviewer comment #1 : The abstract should be rewritten to represent the previous work and why the authors studied this topic.

Author Comment: I have changed the abstract in order to include the state of the art and the proposed goals of this paper.

Reviewer comment #2: Quoting a set of publications (e.g. “  [9], [13],[14]” . "[1], [2],[7] ......[18]"  In this form is unacceptable. Author should analyze the literature more selectively and indicate the most important publications in the analyzed issue together with a commentary.

Author Comment: The introduction was rewritten in order to present the connection between previous recent works (see the references [4]], [11]],[15]) and the current manuscript.

Reviewer comment #3 Highlight some applications of the proposed contraction.

Author comment:  An application to the proposed generalized contraction was given in Example 4. In addition, the example #2 was added in order to show that the dislocated metric spaces over topological modules are not cone metric  spaces.

Reviewer comment #4:  Concluding points can be made more focusing.

Author comment: The conclusion section was changed according to reviewer suggestion and there was included a briefly summary of results and further work  

Reviewer comment #5: No recent citations and references are being cited in this manuscript and this is not  acceptable.

Author comment: There were removed  some references which are not strongly linked to the studied subject and there were added  the new references [4], [11], [15] .

Reviewer 2 Report

The author in this paper provided a generalized contraction-type mapping,
which is defined on a vectorial dislocated space over a topological left module. He present a fixed-point theorem, that generalizes the available results, improving the current work in the literature.

The results seem to be interested and correct. The paper is well writing and organized. There are some non-effected typos can easily improve.

I recommend to publish in Axioms.

Author Response

Dear reviewer,

Thank you for your review. I am delighted that my work is considered as interesting and useful for  the topic of fixed point theory.  Also I have considered your recommendation related to English typos.

Reviewer 3 Report

The paper is very interesting and provides a useful contribution to its area of research.

After consideration of all the above I completely endorse the paper for publication in the journal. The paper can be published in its present form.

Author Response

Dear reviewer,

Thank you for your review. I am delighted that my work is considered  interesting and useful for  the topic of fixed point theory.

Reviewer 4 Report

1. The contribution and novelty can be made clearer in the introduction. In the present form, only some works and backgrounds have been mentioned. 

2. In the first paragraph of the second section, you could mention On r-noncommuting graph of finite rings. This is recent work and shows the relevance to the journal Axioms. 

3. What is the difference between the topological left R module and the right R module? The latter has not been defined in the paper. 

4. The proof of Lemma 1(iii) is too brief. I suggest a complete proof given the length of the paper is short. In general, this applies to other parts of the paper as Axioms has a wide audience with different backgrounds. It is essential to increase accessibility by giving sufficient details. 

5. Example 1 is a straightforward one. Are there any other more sophisticated examples or potential applications?

6. Remark 2 needs some more explanations. It is not easy to follow.

7. The three hypotheses are interesting points. H1 is natural. However, H2 and H3 need some motivation. The definition of Hausdorff space should also be given in the current setting.

8. Third line below equation (12): the inequality needs some explanation.

9. Line 171, via condition C2 we find that there exits an element z0 such that dn tends to z0. This is not immediately clear.

10. The conclusion is too short. Some future directions should be briefly mentioned.

11. A general concern is relevance to Axioms. The author should explain why this paper is suitable for Axioms and emphasis the concept.

Author Response

Reviewer comment #1: The contribution and novelty can be made clearer in the introduction. In the present form, only some works and backgrounds have been mentioned.

Author comment: The introduction was rewritten in order to present the connection between previous recent works (see the references [4]], [11]],[15]) and the current manuscript. Also the goals of the current manuscript was highlighted .

Reviewer comment #2: In the first paragraph of the second section, you could mention On r-noncommuting graph of finite rings. This is recent work and shows the relevance to the journal Axioms. 

Author comment: I have used the above mentioned paper as a guide for my changes but I did not add it to the reference as it is not related to the fixed point theory.

Reviewer comment #3 What is the difference between the topological left R module and the right R module? The latter has not been defined in the paper.

Author comment: I have added the definitions 3 and 4.  

Reviewer comment #4:The proof of Lemma 1(iii) is too brief. I suggest a complete proof given the length of the paper is short. In general, this applies to other parts of the paper as Axioms has a wide audience with different backgrounds. It is essential to increase accessibility by giving sufficient details. 

Author comment: I have included the requested proof  and details to ensure that the paper can be easily understood by general audience

Reviewer comment #5: Example 1 is a straightforward one. Are there any other more sophisticated examples or potential applications?

Author comment:  The example # 2 was added in order to show that the dislocated metric spaces over topological modules are not cone metric  spaces.

Reviewer comment #6:. Remark 2 needs some more explanations. It is not easy to follow.

Author comment: The proof for Remark 2 was added

Reviewer comment #7: The three hypotheses are interesting points. H1 is natural. However, H2 and H3 need some motivation. The definition of Hausdorff space should also be given in the current setting.

Author comment: I have added a comment after Example 3  where I mentioned the reason for considering of hypotheses 2 and 3. Also I mentioned in the  hypothesis H_3 what a Hausdorff space means.

Reviewer comment #8: Third line below equation (12): the inequality needs some explanation.

Author comment: The requested explanations were added on the page 7 last three lines 

Reviewer comment #9. Line 171, via condition C2 we find that there exits an element z0 such that dn tends to z0. This is not immediately clear.

Author comment: The above affirmation was explained on the page 9 line 199

Reviewer comment #10: The conclusion is too short. Some future directions should be briefly mentioned.

Author comment:  The conclusion section was changed according to reviewer suggestion and there was included a briefly summary of results and further work  

Reviewer comment #11: A general concern is relevance to Axioms. The author should explain why this paper is suitable for Axioms and emphasis the concept.

Author comment:   I have modified the abstract, Introduction and conclusion in order to highlight the goals of the paper, the connection with the  recent works in the field of generalized contraction and   the new concepts introduced by this paper (A Cauchy sequence, vectorial dislocated metric space, generalized contraction on vectorial dislocated metric spaces, the strictly inclusion between cone metric space and dislocated metric spaces.)

Round 2

Reviewer 1 Report

The author had changed the abstract in order to include the state of the art and the proposed goals of this paper as required, But it is not acceptable to me to find two equations in the abstract. I think it must be rewritten , otherwise the Editor of the Journal accept this issue.

Author Response

Dear reviewer,

Thank you for your valuable review. I have updated the abstract section, as you suggested, by removing both equations.

Yours sincerely,

Dr. Ion Marian Olaru

Reviewer 4 Report

All comments have been addressed. I recommend it for publication in its current form. 

Author Response

Dear reviewer,

Thank you for your valuable review.

Yours sincerely,

Dr. Ion Marian Olaru